# Oracle-Efficient Reinforcement Learning
# for Max Value Ensembles

**Marcel Hussing**
Dept. of Computer and Information Science
University of Pennsylvania
Philadelphia, PA 19104
mhussing@seas.upenn.edu

**Michael Kearns**
Dept. of Computer and Information Science
University of Pennsylvania
Philadelphia, PA 19104
mkearns@cis.upenn.edu

**Aaron Roth**
Dept. of Computer and Information Science
University of Pennsylvania
Philadelphia, PA 19104
aaroth@cis.upenn.edu

**Sikata Bela Sengupta**
Dept. of Computer and Information Science
University of Pennsylvania
Philadelphia, PA 19104
sikata@seas.upenn.edu

**Jessica Sorrell**[*]
Dept. of Computer Science
Johns Hopkins University
Baltimore, MD 21218
jess@jhu.edu

## Abstract

Reinforcement learning (RL) in large or infinite state spaces is notoriously challenging, both theoretically (where worst-case sample and computational complexities must scale with state space cardinality) and experimentally (where function approximation and policy gradient techniques often scale poorly and suffer from instability and high variance). One line of research attempting to address these difficulties makes the natural assumption that we are given a collection of base or *constituent* policies (possibly heuristic) upon which we would like to improve in a scalable manner. In this work we aim to compete with the *max-following policy*, which at each state follows the action of whichever constituent policy has the highest value. The max-following policy is always at least as good as the best constituent policy, and may be considerably better. Our main result is an efficient algorithm that learns to compete with the max-following policy, given only access to the constituent policies (but not their value functions). In contrast to prior work in similar settings, our theoretical results require only the minimal assumption of an ERM oracle for value function approximation for the constituent policies (and not the global optimal policy or the max-following policy itself) on samplable distributions. We illustrate our algorithm's experimental effectiveness and behavior on several robotic simulation testbeds.

## 1 Introduction

Computationally efficient RL algorithms are known for simple environments with small state spaces such as tabular Markov decision processes (MDPs) [Kearns and Singh, 2002, Brafman and Tennenholtz, 2002], but practical applications often require dealing with large or even infinite state spaces.

---

[*]This work was completed while this author was at the University of Pennsylvania.

38th Conference on Neural Information Processing Systems (NeurIPS 2024).

Learning *efficiently* in these cases requires computational complexity independent of the state space, but this is statistically impossible without strong assumptions on the class of MDPs [Jaksch et al., 2010, Lattimore and Hutter, 2012, Du et al., 2019, Domingues et al., 2021]. Even in structured MDPs that admit statistically efficient algorithms, learning an optimal policy can still be computationally intractable [Kane et al., 2022, Golowich et al., 2024].

These obstacles to practical RL motivate the study of ensembling methods [Lee et al., 2021, Peer et al., 2021, Chen et al., 2021, Hiraoka et al., 2022], which assume access to multiple sub-optimal policies for the same MDP and aim to leverage these constituent policies to improve upon them. There are now several provably efficient ensembling algorithms, but their guarantees require strong assumptions on the representation of the target policy learned by the algorithm. Brukhim et al. [2022] use the boosting framework for ensembling developed in the supervised learning setting [Freund and Schapire, 1997] to learn an optimal policy, assuming access to a weak learner for a parameterized policy class. To efficiently converge to an optimal policy, the target policy must be expressible as a depth-two circuit over policies from a base class which is efficiently weak-learnable. The convergence guarantees additionally require strong bounds on the worst-case distance between state-visitation distributions of the target policy and policies from the base class.

Another line of ensembling work considers a weaker objective than learning an optimal policy [Cheng et al., 2020, Liu et al., 2023, 2024]. These works instead aim to learn a policy competitive with a *max-aggregation policy*, which take whichever action maximizes the advantage function with respect to a max-following policy at the current state. When these works have provable guarantees, they require the assumption that the target max-aggregation policy can be approximated in an online-learnable parametric class, as well as the assumption that policy gradients within the class can be efficiently estimated with low variance and bias.

Our goal is to learn a policy competitive with a similar but incomparable benchmark to that of Cheng et al. [2020] under comparatively weak assumptions. We give an efficient algorithm for learning a policy competitive with a *max-following policy* (Definition 2.1), assuming the learner has access to a squared-error regression oracle for the value functions of the constituent policies. Our algorithm exclusively queries this oracle on distributions over states that are efficiently samplable, thereby reducing the problem of learning a max-following competitive policy to supervised learning of value functions. Notably, our learnability assumptions pertain only to the value functions of the constituent policies and not to the more complicated class of max-following benchmark policies or their value functions. Our algorithm is simple and effective, which we demonstrate empirically in Section 5.

It is natural to wonder if access to an oracle such as ours could be leveraged to instead efficiently learn an optimal policy, obviating the need for weaker benchmarks (and our results). However, it was recently shown by [Golowich et al., 2024] that learning an optimal policy in a particular family of block MDPs is computationally intractable under reasonable cryptographic assumptions, even when the learner has access to a squared-error regression oracle. Their oracle captures a general class of regression tasks that includes value function estimation, and therefore also captures our oracle assumption. Our work shows that when we instead consider the simpler objective of efficiently learning a policy that competes with max-following, a regression oracle is in fact sufficient. We leave open the interesting question of whether such an oracle is necessary.

## 1.1 Results

Our main contribution is a novel algorithm for improving upon a set of $K$ given policies that is oracle efficient with respect to a squared-error regression oracle, and therefore scalable in large state spaces (Algorithm 1, Theorem 3.1). We consider the episodic RL setting in which the learner interacts with its environment for episodes of a fixed length $H$. The algorithm incrementally constructs an improved policy over $H$ iterations, learning an improved policy for step $h \in [H]$ of the episode at iteration $h$. This incremental approach allows the algorithm to explicitly construct efficiently samplable distributions over states visited by the improved policy at step $h$ by simply executing the current policy for $h$ steps. It can then query its oracle to obtain approximate value functions for all constituent policies with respect to this distribution. This in turn allows the algorithm to learn an improved policy for step $h+1$ by following the policy with highest estimated value. By incrementally constructing an improved policy over steps of the episode, we can avoid making assumptions like those of Brukhim et al. [2022] about the overlap between state-visitation distributions of the target policy and the intermediate policies constructed by the algorithm.

As our oracle only gives us approximate value functions, we take as our benchmark class the set of *approximate max-following policies* (Definition 2.3). This is a superset of the class of max-following policies and contains all policies that at each state follow the action of some constituent policy with near-maximum value. In Section 4, we prove that for any set of constituent policies, the worst approximate max-following policy is competitive with the best constituent policy (Lemma 4.1) and provide several example MDPs illustrating how our benchmark relates to other natural benchmarks.

Finally, we demonstrate the practical feasibility of our algorithm using a heuristic version on a set of robotic manipulation tasks from the CompoSuite benchmark Mendez et al. [2022], Hussing et al. [2024]. We demonstrate that in all cases, the max-following policy we find is at least as good as the constituent policies and in several cases outperforms it significantly.

## 1.2 Related work

Our work is related to a recent line of research learning a max-aggregation policy [Cheng et al., 2020, Liu et al., 2023, 2024], which can be viewed as a one-step look-ahead max-following policy and is incomparable to the class of max-following policies (see Cheng et al. [2020] for example MDPs demonstrating this fact). Sekhari et al. [2024] consider the problem of imitation learning from multiple noisy experts using selective sampling. For queried experts, their algorithm invokes an online regression oracle assumption and they leave as an open direction learning with offline regression oracles. These works all assume online learnability of the target policy class, which is strictly stronger than our batch learnability assumption for constituent policy value functions.

The work of Cheng et al. [2020] proposes an algorithm (MAMBA) that uses policy gradient methods, and the convergence of the learned policy to their benchmark depends on the bias and variance of those policy gradients. Liu et al. [2023, 2024] builds on the work of [Cheng et al., 2020]. Their algorithm MAPS-SE modifies MAMBA to promote exploration when there is uncertainty about which constituent policy has the greatest value at a state, via an upper confidence bound (UCB) approach to policy selection. Reducing uncertainty about the constituent policies' value functions reduces the bias and variance of the gradient estimates, improving convergence guarantees. However, policy gradient techniques are known to generally have high variance [Wu et al., 2018], and this appears to affect the practical performance of MAPS-SE in certain cases (see Section 5 for additional discussion).

The boosting approach to policy ensembling of Brukhim et al. [2022] also necessitates very strong assumptions. This follows from the computational separation in Golowich et al. [2024], which shows that our oracle assumption is insufficient to learn an optimal policy, whereas the assumptions made in Brukhim et al. [2022] enable convergence to optimality.

Much work on policy improvement considers improving upon a single base policy and do not address the challenge of ensembling [Sun et al., 2017, Schulman et al., 2015, Chang et al., 2015]. Barreto et al. [2017, 2020], Alegre et al. [2024] consider the problem of Generalized Policy Improvement (GPI) by decomposing complex tasks into a set of multiple smaller tasks where they use transfer learning. However, they make strong assumptions about the joint representation of rewards (tasks) as linear in successor feature representations, which may be challenging to explicitly learn in MDPs that are not tabular. Zaki et al. [2022] consider the setting of access to $M$ base controllers with the aim of optimally combining them to produce a controller that is competitive with the base set. They approach this problem with the aim of considering a *single* controller from the softmax policy class over the base set of policies that is competitive with all the others, but not in a *state-dependent* manner. Empirical work on ensemble imitation learning (IL) also studies the problem of leveraging multiple base policies for learning [Li et al., 2018, Kurenkov et al., 2019], but these works lack provable guarantees of efficient convergence to a meaningful benchmark. [Song et al., 2023] provide a survey of a variety of more complex techniques to ensemble policies, mainly from a practical perspective.

## 2 Preliminaries

We consider an episodic fixed-horizon Markov decision process (MDP) [Puterman, 1994] which we formalize as a tuple $\mathcal{M} = (\mathcal{S}, \mathcal{A}, R, P, \mu_0, H)$ where $\mathcal{S}$ is the set of states, $\mathcal{A}$ the set of actions, $R$ is a reward function, $P$ the transition dynamics, $\mu_0$ a distribution over starting states and $H$ the horizon [Sutton and Barto, 2018]. $[N]$ will denote the set $\{0, ..., N-1\}$. In the beginning, an initial state $s_0$

is sampled from $\mu_0$. At any time $h \in [H]$, the agent is in some state $s_h \in \mathcal{S}$ and chooses an action $a_h \in \mathcal{A}$ based on a function $\pi_h$ mapping from states to distributions over actions $\Pi : \mathcal{S} \mapsto \Delta(\mathcal{A})$. As a consequence, the agent traverses to a new next state $s_{h+1}$ sampled from $P(\cdot|s_h, a_h)$ and obtains a reward $R(s_h, a_h)$. Without loss of generality, we assume that rewards bounded within $[0, 1]$. The sequence of functions $\pi_h$ used by the agent is referred to as its *policy*, and is denoted $\pi = \{\pi_h\}_{h \in [H]}$. A *trajectory* is the sequence of (state, action) pairs taken by the agent over an episode of length $H$, and is denoted $\tau = \{(s_h, a_h)\}_{h \in [H]}$. We will use the notation $\tau \sim \pi(\mu_0)$ to refer to sampling a trajectory by first sampling a starting state $s_0 \sim \mu_0$, and then executing policy $\pi$ from $s_0$.

The goal of the learner is to maximize the expected cumulative reward $\mathbb{E}_{s_0 \sim \mu_0, P}[\sum_{t=0}^{H-1} R(s_t, a_t)]$ over episodes of length $H$. We further define the value function as the expected cumulative return of following some policy $\pi$ from some state $s$ as $V^\pi(s) = \mathbb{E}_{s_0 \sim \mu_0, P}[\sum_{t=0}^{H-1} R(s_t, a_t)|\pi, s_0 = s]$. Due to the finite horizon of the episodic setting, we will also need to refer to the expected cumulative reward from state $s$ under policy $\pi$ from time $h \in [H]$. We denote this time-specific value function by $V_h^\pi(s) = \mathbb{E}_P[\sum_{t=h}^{H-1} R(s_t, a_t)|\pi, s_h = s]$. Finally, the key object of interest is a max-following policy. Given access to a set of $k$ arbitrarily defined policies $\Pi^k = \{\pi^k\}_{k=1}^K$ and their respective value functions which we denote by the shorthand $V^{\pi_k} = V^k$, a max-following policy is defined as a policy that at every step follows the action of the policy with the highest value in that state.

**Definition 2.1** (Max-following policy class). *Fix a set of policies $\Pi^k$ for a common MDP $\mathcal{M}$ and an episode length $H$. The class of* max-following *policies $\Pi_{\max}^k$ is defined*

$$\Pi_{\max}^k = \{\pi : \forall h \in [H], \forall s \in \mathcal{S}, \pi_h(s) = \pi^{k^*}(s) \, \text{for some } k^* \in \underset{k \in [K]}{\arg\max} \, V_h^k(s)\}$$

Note that for any collection of constituent policies $\Pi^k$ there may be many max-following policies, due to ties between the value functions. Different max-following policies may have different expected return, and we refer the reader to Observation 4.5 for an example demonstrating this fact.

We assume access to a value function oracle that allows us to approximate a value function of a policy under a samplable distribution at any specified time $h \in [H]$. This oracle is intended to capture the common assumption that the value function of a policy can be efficiently well-approximated by a function from a fixed parameterized class. In practice, one might imagine implementing this oracle as a neural network minimizing the squared error to a target value function.

**Definition 2.2** (Oracle for $\pi$ value function estimates). *We denote by $\mathcal{O}^\pi$ an oracle satisfying the following guarantee for a policy $\pi$. For any $\alpha \in (0, 1]$, and any $h \in [H]$, given as input a time $h \in [H]$ and sampling access to any efficiently samplable distribution $\mu$, the oracle outputs $\hat{V}_h^\pi \leftarrow \mathcal{O}^\pi(\alpha, \mu, h)$ such that $\mathbb{E}_{s \sim \mu}[(\hat{V}_h^\pi(s) - V_h^\pi(s))^2] \leq \alpha$. We use the notation $\mathcal{O}_\alpha^\pi = \mathcal{O}^\pi(\alpha, \cdot, \cdot)$ to denote $\mathcal{O}^\pi$ with fixed accuracy parameter $\alpha$. We will also use the shorthand $\mathcal{O}^k = \mathcal{O}^{\pi^k}$.*

Looking ahead to Section 3, we note that for every distribution $\mu$ on which Algorithm 1 queries an oracle, $\mu$ is not only efficiently samplable, but samplable by executing an explicitly constructed policy $\pi_{\text{samp}}$ for $h$ steps in MDP $\mathcal{M}$, starting from $\mu_0$. Thus, for any distribution $\mu$, policy $\pi^k$, and time $h$ for which we query $\mathcal{O}^k$, we could efficiently obtain an unbiased estimate of $\mathbb{E}_{s \sim \mu}[V_h^k(s)]$ by following a known $\pi_{\text{samp}}$ for $h$ steps from $\mu_0$, and then switching to $\pi^k$ for the remainder of the episode. We mention this to highlight that our oracle is not eliding any technical obstacles to sampling in the episodic setting. It is simply abstracting the supervised learning task of converting unbiased estimates of $\mathbb{E}_{s \sim \mu}[V_h^k(s)]$ into an approximation $\hat{V}_h^k$ with small squared error with respect to $\mu$.

Lastly, we define our benchmark class of policies. Given a set of constituent policies $\Pi^k$, our benchmark defines for each state and time a set of permissible actions: any action taken by a policy $\pi^t \in \Pi^k$ for which the value $V_h^t(s)$ is sufficiently close to the maximum value $\max_{k \in [K]} V_h^k(s)$. The class of approximate max-following policies is then any policy that exclusively takes permissible actions. We refer the reader to Section 4 for further explanation of this benchmark.

**Definition 2.3** (Approximate max-following policies). *We define a set of $\beta$-good policies at state $s \in \mathcal{S}$ and time $h \in [H]$, selected from a set $\Pi^k$, as follows.*

$$T_{\beta, h}(s) = \{\pi \in \Pi^k : V_h^\pi(s) \geq \max_{k \in [K]} V_h^k(s) - \beta\}.$$

*Then we define the set of approximate max-following policies for $\Pi^k$ to be*

$$\Pi_\beta^{k^*} = \{\pi : \forall h \in [H], \forall s \in \mathcal{S}, \pi_h(s) = \pi_h^t(s) \, \text{for some } \pi^t \in T_{\beta, h}(s)\}.$$

# 3 The MaxIteration **learning algorithm**

In this section, we introduce our algorithm for learning an approximate max-following policy, MaxIteration (Algorithm 1. This algorithm learns a good approximation of a max-following policy at step $h$, assuming access to a good approximation of a max-following policy for all previous steps.

For the first step ($h = 0$), the algorithm learns a good approximation $\hat{V}_0^k$ for all constituent policies $\pi^k$ on the starting distribution $\mu_0$. These approximate value functions can in turn be used to define the first action taken by the approximate max-following policy, namely $\hat{\pi}_0(s) = \pi_{\mathrm{argmax}_k \hat{V}_0^k(s)}(s)$. Following $\hat{\pi}_0(s)$ from $\mu_0$ generates a samplable distribution over states $\mu_1(s) = \mathbb{E}_{s_0 \sim \mu_0}[P(s|s_0, \hat{\pi}_0(s_0))]$, and so our oracle assumption allows us to obtain good estimates $\hat{V}_1^k$ with respect to $\mu_1$ for all $\pi^k$. We can then define the second action of the approximate max-following policy, and so on, for all $H$ steps.

Notice that sampling from $\mu_h$ does not require that the agent can reset the environment at will. It only requires what is typically required in the episodic setting – that the agent explores for an episode of $H$ steps, where $H$ is finite and fixed across all of training. After these $H$ steps, the agent is then reset to a state sampled from the distribution over starting states. The distributions $\mu_h$ are (informally) defined as follows: at iteration $h \in [H]$ of our algorithm, the agent has already learned a good approximate max-following policy for the first $h$ steps of the episode. The distribution $\mu_h$ is the distribution over states visited by the agent at step $h$ if it begins from a state drawn from the starting state distribution and then follows the approximate max-following policy it has learned thus far for $h$ steps. That means to sample from $\mu_h$, the oracle can simply run the approximate max-following policy for $h$ steps to arrive at a state $s_h$, which is a sample from $\mu_h$. It can then do anything for the remainder of the episode, and so does not need to reset at arbitrary time steps. In practice, since the oracle needs to produce a good approximation of the value function $V_h^k$ at time $h$ for policy $\pi^k$ on states sampled from $\mu_h$, one should think of it as using the remainder of the episode to obtain an unbiased estimate of the expectation of $V_h^k$ on the distribution $\mu_h$. That is, once it has sampled a state $s_h$ by running the approximate max-following policy for $h$ steps, it just executes policy $\pi^k$ for the remainder of the episode. The accumulated reward obtained by following policy $\pi^k$ from state $s_h$ for steps $h$ through $H$ gives the oracle an unbiased estimate of $\mathbb{E}_{s_h \sim \mu_h}[V_h^k(s_h)]$. To implement this oracle assumption, one could use many such unbiased estimates as training data to train a neural network, to learn a good approximate value function for $\pi^k$ at time $h$ on distribution $\mu_h$.

---

**Algorithm 1** MaxIteration$_\alpha^{\mathcal{M}}(\Pi^k)$

---

1: **for** $h \in [H]$ **do**
2:     **for** $k \in [K]$ **do**
3:        let $\mu_h$ be the distribution sampled by executing the following procedure:
4:           sample a starting state $s_0 \sim \mu_0$
5:           **for** $i \in [h]$ **do**
6:              $s_{i+1} \sim P\big( \cdot \mid s_i, \pi^{\mathrm{argmax}_k \hat{V}_i^k(s_i)}(s_i)\big)$
7:           **end for**
8:           output $s_h$
9:        $\hat{V}_h^k \leftarrow \mathcal{O}_\alpha^k(\mu_h, h)$
10:     **end for**
11: **end for**
12: return policy $\hat{\pi} = \{\hat{\pi}_h\}_{h \in [H]}$ where $\hat{\pi}_h(s) = \pi^{\mathrm{argmax}_{k \in [K]} \hat{V}_h^k(s)}(s)$

---

**Theorem 3.1.** *For any $\varepsilon \in (0, 1]$, any MDP $\mathcal{M}$ with starting state distribution $\mu_0$, any episode length $H$, and any $K$ policies $\Pi^k$ defined on $\mathcal{M}$, let $\alpha \in \Theta(\frac{\varepsilon^3}{KH^4})$ and $\beta \in \Theta(\frac{\varepsilon}{H})$. Then* MaxIteration$_\alpha^{\mathcal{M}}(\Pi^k)$ *makes $O(HK)$ oracle queries and outputs $\hat{\pi}$ such that*

$$\mathbb{E}_{s_0 \sim \mu_0} \big[ V^{\hat{\pi}}(s_0) \big] \geq \min_{\pi \in \Pi_\beta^{k*}} \mathbb{E}_{s_0 \sim \mu_0} \big[ V^\pi(s_0) \big] - O(\varepsilon).$$

*Proof.* For all $h \in [H]$, $k \in [K]$, let $\hat{V}_h^k$ denote the approximate value function obtained from $\mathcal{O}_\alpha^k(\mu_h, h)$ in Algorithm 1. We then define, for every $h \in [H]$, the set of states for which some approximate value function $\hat{V}_h^k(s)$ has large absolute error ($B_h$) and the set of bad trajectories ($B_\tau$) that

pass through a state in $B_h$ for any $h \in [H] : B_h = \{s \in S : \exists k \in [K] \text{ s.t. } |\hat{V}_h^k(s) - V_h^k(s)| \geq \frac{\varepsilon}{2H}\}$ and $B_\tau = \{\{(s_h, a_h)\}_{h \in [H]} : \exists h \in [H] \text{ s.t. } s_h \in B_h\}$. We will show that there exists an approximate max-following policy $\pi \in \Pi_\beta^{k^*}$ such that for any trajectory $\tau' \notin B_\tau$, $\text{Pr}_{\tau \sim \hat{\pi}(\mu_0)}[\tau = \tau'] = \text{Pr}_{\tau \sim \pi(\mu_0)}[\tau = \tau']$. We then bound the probability $\text{Pr}_{\tau \sim \hat{\pi}(\mu_0)}[\tau \in B_\tau]$, and the contribution to $\mathbb{E}_{s_0 \sim \mu_0}[V^\pi(s_0)]$ from these trajectories, proving the claim.

Let $V_h^{k^*}(s)$ denote the value of the policy that $\hat{\pi}$ follows at time $h$ and state $s$. From the definition of the bad set $B_h$ and the setting of $\beta \in \Theta(\frac{\varepsilon}{H})$, for any state $s \notin B_h$,

$$V_h^{k^*}(s) \geq \hat{V}_h^{k^*}(s) - \frac{\varepsilon}{2H} \geq \max_{k \in [K]} \hat{V}_h^k(s) - \frac{\varepsilon}{2H} \geq \max_{k \in [K]} V_h^k(s) - \beta.$$

In other words, if a state $s$ is not bad at time $h$, then $\hat{\pi}_h(s) = \pi_h^k(s)$ for a policy $\pi^k$ that has value $V_h^k(s)$ within $\beta$ of the true max value $\max_{k \in [K]} V_h^k(s)$. It then follows from the definition of the class of approximate max-following policies $\Pi_\beta^{k^*}$ (Definition 2.3) that there exists some $\pi \in \Pi_\beta^{k^*}$ such that for all $h \in [H]$, for all $s \notin B_h$, $\hat{\pi}_h(s) = \pi_h(s)$.

For any trajectory $\tau'$, $\text{Pr}_{\tau \sim \hat{\pi}(\mu_0)}[\tau = \tau'] = \text{Pr}_{\mu_0}[s_0] \cdot \prod_{h=0}^{H-1} P(s_{h+1}|s_h, \hat{\pi}_h(s_h))$. Then for any trajectory $\tau' \notin B_\tau$, $\text{Pr}_{\tau \sim \hat{\pi}(\mu_0)}[\tau = \tau'] = \text{Pr}_{\tau \sim \pi(\mu_0)}[\tau = \tau']$, and therefore

$$\mathbb{E}_{\tau \sim \hat{\pi}(\mu_0)}\left[\sum_{h=0}^{H-1} R(s_h, a_h) \mid \tau \notin B_\tau\right] = \mathbb{E}_{\tau \sim \pi(\mu_0)}\left[\sum_{h=0}^{H-1} R(s_h, a_h) \mid \tau \notin B_\tau\right]$$

For $\tau \in B_\tau$, we have lower and upper-bounds $\mathbb{E}_{\tau \sim \hat{\pi}(\mu_0)}[\sum_{h=0}^{H-1} R(s_h, a_h) \mid \tau \in B_\tau] \geq 0$ and $\mathbb{E}_{\tau \sim \pi(\mu_0)}[\sum_{h=0}^{H-1} R(s_h, a_h) \mid \tau \in B_\tau] \leq H$. We can then write:

$$\mathbb{E}_{s_0 \sim \mu_0}\left[V^{\hat{\pi}}(s_0)\right] = \mathbb{E}_{\tau \sim \hat{\pi}(\mu_0)}\left[\sum_{h=0}^{H-1} R(s_h, a_h) \mid \tau \notin B_\tau\right] \cdot \text{Pr}_{\tau \sim \hat{\pi}(\mu_0)}[\tau \notin B_\tau]$$

$$+ \mathbb{E}_{\tau \sim \hat{\pi}(\mu_0)}\left[\sum_{h=0}^{H-1} R(s_h, a_h) \mid \tau \in B_\tau\right] \cdot \text{Pr}_{\tau \sim \hat{\pi}(\mu_0)}[\tau \in B_\tau]$$

$$\geq \mathbb{E}_{\tau \sim \hat{\pi}(\mu_0)}\left[\sum_{h=0}^{H-1} R(s_h, a_h) \mid \tau \notin B_\tau\right] \cdot \text{Pr}_{\tau \sim \hat{\pi}(\mu_0)}[\tau \notin B_\tau]$$

$$= \mathbb{E}_{\tau \sim \pi(\mu_0)}\left[\sum_{h=0}^{H-1} R(s_h, a_h) \mid \tau \notin B_\tau\right] \cdot \text{Pr}_{\tau \sim \pi(\mu_0)}[\tau \notin B_\tau]$$

$$\geq \mathbb{E}_{\tau \sim \pi(\mu_0)}\left[\sum_{h=0}^{H-1} R(s_h, a_h)\right] - H \cdot \text{Pr}_{\tau \sim \pi(\mu_0)}[\tau \in B_\tau] \quad \text{(using law of total probability and upper bound on rewards)}$$

$$\geq \min_{\pi \in \Pi_\beta^{k^*}} \mathbb{E}_{s_0 \sim \mu_0}[V^\pi(s_0)] - H \cdot \text{Pr}_{\tau \sim \pi(\mu_0)}[\tau \in B_\tau].$$

It remains to upper-bound $\text{Pr}_{\tau \sim \pi(\mu_0)}[\tau \in B_\tau]$. We have already argued $\text{Pr}_{\tau \sim \pi(\mu_0)}[\tau \in B_\tau] = \text{Pr}_{\tau \sim \hat{\pi}(\mu_0)}[\tau \in B_\tau]$. Observing that $\text{Pr}_{\tau \sim \hat{\pi}(\mu_0)}[\tau \in B_\tau] \leq \sum_{h=0}^{H-1} \text{Pr}_{\tau \sim \hat{\pi}(\mu_0)}[s_h \in B_h]$, it is sufficient to show $\text{Pr}_{\tau \sim \hat{\pi}(\mu_0)}[s_h \in B_h] \in O(\frac{\varepsilon}{H^2})$ to prove the claim. For all $h \in [H]$, let $\mu_h(s) = \text{Pr}_{\tau \sim \hat{\pi}(\mu_0)}[s_h = s]$, and note that this is the distribution supplied to the oracle at iteration $h$ of Algorithm 1. It follows from our oracle assumption (Definition 2.2) that for all $k \in [K]$, $\mathbb{E}_{s_h \sim \mu_h}[(\hat{V}^k(s_h) - V^k(s_h))^2] < \alpha$. We apply Markov's inequality to conclude that for all $k \in [K]$,

$$\text{Pr}_{s_h \sim \mu_h}[|\hat{V}_h^k(s_h) - V_h^k(s_h)| \geq \frac{\varepsilon}{2H}] < \frac{4\alpha H^2}{\varepsilon^2} \in O(\frac{\varepsilon}{KH^2}).$$

Union bounding over the $K$ constituent policies gives $\text{Pr}_{s_h \sim \mu_h}[s_h \in B_h] \in O(\frac{\varepsilon}{H^2})$, from the definition of $B_h$. Union bounding over the trajectory length $H$, we then have $\text{Pr}_{\tau \sim \hat{\pi}(\mu_0)}[\tau \in B_\tau] \in O(\frac{\varepsilon}{H})$. It follows that

$$\mathbb{E}_{s_0 \sim \mu_0}\left[V^{\hat{\pi}}(s_0)\right] \geq \min_{\pi \in \Pi_\beta^{k^*}} \mathbb{E}_{s_0 \sim \mu_0}[V^\pi(s_0)] - O(\varepsilon),$$

completing the proof. $\square$

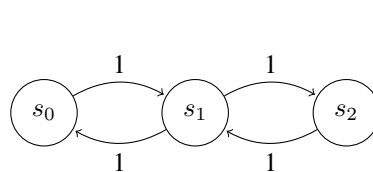

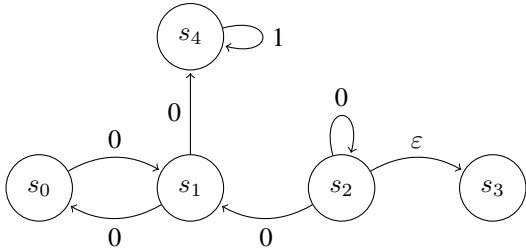

(a) MDP in which two policies going either only left or right obtain low return but max-following them would be optimal.

(b) MDP with $\mathcal{A} = \{\mathsf{right}, \mathsf{left}, \mathsf{up}\}$ where starting from $s_2$, max-following is far worse than optimal and starting from $s_0$, different max-following policies have different values (depending on tie-breaking).

Figure 1: Examples of MDPs with max-following policy performance comparison

## 4  The approximate max-following benchmark

In this section, we provide additional context for our benchmark class of approximate max-following policies. We show that the worst policy in our benchmark class competes with the best fixed policy from the set of constituent policies. We also provide examples of MDPs that showcase properties of the set of (approximate) max-following policies.

**Lemma 4.1** (Worst approximate max-following policy competes with best fixed policy)**.** *For any* $\varepsilon \in (0,1]$ *and any episode length* $H$*, let* $\beta \in \Theta(\frac{\varepsilon}{H})$*. Then for any MDP* $\mathcal{M}$ *with starting state distribution* $\mu_0$*, and any* $K$ *policies* $\Pi^k$ *defined on* $\mathcal{M}$*,*

$$\min_{\pi \in \Pi_\beta^{k*}} \mathbb{E}_{s_0 \sim \mu_0} \left[ V^{\hat{\pi}}(s_0) \right] \geq \max_{k \in [K]} \mathbb{E}_{s_0 \sim \mu_0} \left[ V^k(s_0) \right] - O(\varepsilon).$$

We defer the proof of Lemma 4.1 to Appendix B.

It is an immediate corollary of Theorem 3.1 and Lemma 4.1 that the policy learned by Algorithm 1 competes with the best constituent policy.

**Corollary 4.2.** *For any* $\varepsilon \in (0,1]$*, any MDP* $\mathcal{M}$ *with starting state distribution* $\mu_0$*, any episode length* $H$*, and any* $K$ *policies* $\Pi^k$ *defined on* $\mathcal{M}$*, let* $\alpha \in \Theta(\frac{\varepsilon^3}{KH^4})$*, and let* $\hat{\pi}$ *denote the policy output by* $\mathsf{MaxIteration}_\alpha^{\mathcal{M}}(\Pi^k)$*. Then*

$$\mathbb{E}_{s_0 \sim \mu_0} \left[ V^{\hat{\pi}}(s_0) \right] \geq \max_{k \in [K]} \mathbb{E}_{s_0 \sim \mu_0} \left[ V^k(s_0) \right] - O(\varepsilon).$$

We provide diagrams of MDPs as examples for the observations that we make below. States in $\mathcal{S}$ are denoted by the labels on the nodes. Actions in $\mathcal{A}$ are indicated by arrows from given states with deterministic transition dynamics and the rewards $R(s,a)$ are labeled over the corresponding arrows. Arrows may be omitted for transitions that are self-loops with reward 0.

**Observation 4.3.** *The worst approximate max-following policy can be arbitrarily better than the best constituent policy.*

Consider in Figure 1a two policies on this MDP: $\pi^0(s) = \mathsf{right}$ and $\pi^1(s) = \mathsf{left}$, for all $s \in \mathcal{S}$. Note that for any episode length $H \geq 2$, for all $k \in \{0,1\}$, $\max_{s \in \mathcal{S}} V^k(s) = 2$. For any $\beta < 1$, $\Pi_\beta^{k*}$ comprises policies $\pi$ such that $\pi(s_0) = \mathsf{right}$, $\pi(s_2) = \mathsf{left}$, and $\pi(s_1) \in \{\mathsf{right}, \mathsf{left}\}$. Therefore for any episode length $H$, and state $s \in \mathcal{S}$, $\min_{\pi \in \Pi_\beta^{k*}} V^\pi(s) = H$. In this example, any approximate max-following policy is also an optimal policy, whose gap in expected return with the best constituent policy can be made arbitrarily large by increasing $H$.

**Observation 4.4.** *A max-following policy cannot always compete with an optimal policy.*

In Figure 1b, consider policies $\pi^0(s) = \mathsf{right}$, $\pi^1(s) = \mathsf{left}$, and $\pi^2(s) = \mathsf{up}$, for all $s \in \mathcal{S}$. At state $s_2$, $\pi^0$ is the only policy with non-zero value. Thus, any max-following policy will take action right from $s_2$, receiving reward $\varepsilon$ and then reward 0 for the remainder of the episode. Given a starting state distribution supported entirely on $s_2$, for any episode length $H \geq 3$, the optimal policy will obtain cumulative reward $H - 2$, whereas any max-following policy will only obtain reward $\varepsilon$.

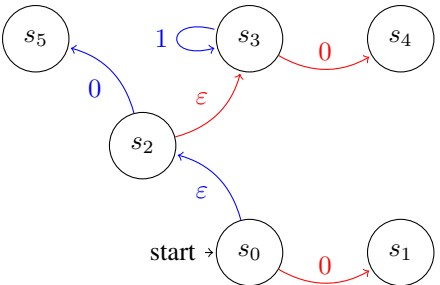
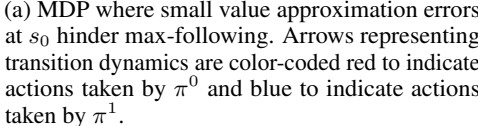
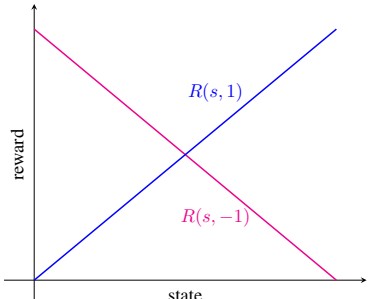

(a) MDP where small value approximation errors at $s_0$ hinder max-following. Arrows representing transition dynamics are color-coded red to indicate actions taken by $\pi^0$ and blue to indicate actions taken by $\pi^1$.

(b) MDP where the max-following value function is piecewise linear, but constituent policy's values are affine functions of the state for fixed actions.

Figure 2: Examples for Observation 4.6 and Observation 4.7

**Observation 4.5.** *Different max-following policies may have different expected cumulative reward.*

We again consider Figure 1b, but suppose now the starting state distribution is supported entirely on $s_0$. For all $k \in [3]$, $V^k(s_0) = 0$ and so a max-following policy may take any action from $s_0$. A max-following policy that always takes actions left or up from $s_0$ will only ever obtain cumulative reward 0, but a max-following policy that takes action right will move to $s_1$ and (so long as more than one step remains in the episode) will then take action up and move to state $s_4$, where it will stay to obtain cumulative reward $H - 2$.

If the value functions of constituent policies are exactly known, it is easy to construct a max-following policy, but the learner may not have access to these functions. If the learner only has access to approximations and follows whichever policy has the larger approximate value at the current state, the resulting policy can have much lower expected cumulative reward than the max-following policy. This is true even for state-wise bounds on the value approximation error. This observation previously motivated our definition of the approximate max-following class (Definition 2.3).

**Observation 4.6.** *Small value function approximation errors can be an obstacle to learning a max-following policy.*

In Figure 2a, we again consider policies $\pi^0(s) =$ right and $\pi^1(s) =$ left for all states $s \in \mathcal{S}$, color coding the actions taken by $\pi^0$ with red and $\pi^1$ with blue in Figure 2a. For starting state distribution supported entirely on $s_0$, a max-following policy $\pi$ will take action $\pi(s_0) =$ left, $\pi(s_2) =$ right, and $\pi(s_3) =$ left for the remainder of the episode, obtaining reward $H - 2 + 2\varepsilon$. However, given only approximate value functions $\hat{V}^k$ with state-wise absolute error bound $|\hat{V}_h^k(s) - V_h^k(s)| \leq \varepsilon$ for all states $s$ and times $h$, the policy $\hat{\pi}$ that takes action $\pi_h^{k^*}(s)$ for $k^* = \mathrm{argmax}_{k \in [2]} \hat{V}_h^k(s)$ can have much lower expected cumulative reward than a max-following policy. For example if $\hat{V}_0^0(s_0) = \varepsilon$ and $\hat{V}_0^1(s_0) = 0$ in our Figure 2a example, then $\hat{\pi}$ will have expected return 0.

**Observation 4.7.** *A max-following policy's value function is not always of the same parametric class as the constituent policies' value functions.*

As a simple first example, consider an MDP with states $\mathcal{S} = [0, 1]$ and actions $\mathcal{A} = \{-1, 1\}$. Every action leads to a self-loop (for all $a \in \mathcal{A}$, $P(s|s, a) = 1$) and for a fixed action, rewards are affine functions of the state (e.g. $R(s, -1) = 1 - s$ and $R(s, 1) = s$). We consider two policies: $\pi^0(s) = -1$ and $\pi^1(s) = 1$ for all $s \in \mathcal{S}$. Notice that for episode length $H$, $V^0(s) = HR(s, -1)$ and $V^1(s) = HR(s, 1)$. Since the dynamics keep the state at the same fixed place independent of the action, the max-following policy at state $s$ will simply be the max of the two individual value functions at $s$ and therefore its parametric class will be piecewise linear, unlike the constituent policies' which are affine (see Figure 2b). To provide a more complex MDP example, we consider a traditional control problem with continuous state and action spaces: the discrete linear quadratic regulator. In this example the constituent linear policies have quadratic value functions, but the max-following policy is not of the same parametric class. See Appendix A for further discussion.

# 5   Experiments

We proceed to examine our MaxIteration algorithm in a set of experiments that uses neural network function approximation as oracles. These experiments aim to provide a scenario to demonstrate the usefulness of max-following. While previous works in this line of research have studied the ability to integrate knowledge from the constituent policies to increase performance of a learnable policy [Cheng et al., 2020, Liu et al., 2023, 2024] our algorithm offers an alternative approach. We consider a common scenario from the field of robotics where one has access to older policies from a robotic simulator that were used in previous projects. As long as the dynamics of the MDP of interest do not differ, such old policies can be simply be re-used in new applications. In such cases, training completely from scratch can be incredibly expensive due to the vast search space [Schulman et al., 2017, Haarnoja et al., 2018]. We note that this setup is related to the one used by Barreto et al. [2017, 2020] but we do not put any constraints on the reward functions.

**Experimental setup**   A recent robotic simulation benchmark called CompoSuite [Mendez et al., 2022] and its corresponding offline datasets [Hussing et al., 2024] offer an instantiation of such a scenario. CompoSuite consists of four axes: robot arms, objects, objectives and obstacles. Tasks are simply constructed by combining one element from each axis.We consider tasks with a fixed IIWA robotic manipulator and no obstacle. This leaves us with a total of 16 tasks. These 16 tasks are randomly grouped into pairs of two. Each group is one experiment where the policies trained on tasks correspond to our constituents. To create a new target task, we change one element per task, creating novel combinations for each group. For example, we start with the constituent policies that can 1) put and place a box into a trashcan and 2) push a plate. The target task can be to push the box. We train our constituent policies on the expert datasets using the offline RL algorithm Implicit Q-learning [Kostrikov et al., 2022] (IQL). This ensures we obtain very strong constituent policies for their respective tasks. After training the constituents, we run MaxIteration and the baselines for a short amount of time in the simulator. We report mean performance and standard error over 5 seeds using an evaluation of 32 episodes.

**Algorithms**   For practical purposes, we use a heuristic version of MaxIteration which does not re-compute the max-following policy at every step $h$ but rather after multiple steps. For our baselines, we ran the code provided by [Liu et al., 2023] to train the MAPS algorithm but were unable to obtain non-trivial return even after a reasonable amount of tuning. MAPS has been shown to have difficulties with leveraging very performant constituent policies such as the ones we are using (see the Walker experiment by Liu et al. [2023] in Figure 1 (d) in which the algorithm struggles to be competitive with the best, high-return constituent policy). They conjecture that in this case, their estimates of the constituent value functions will be less accurate in early training, resulting in gradient estimates with large bias and variance, weakening their convergence guarantees. We provide an evaluation of MaxIteration on tasks originally used by Liu et al. [2023] in Appendix C.3.

For now, we opt to use IQL's in fine-tuning capabilities that offer a policy improvement style method on top of the best-performing constituent policy for comparison. Fine-tuning provides a strong baseline in the sense that it has access to the already trained value functions of the constituent policies providing it with inherently more starting information. For comparability, we limit the number of episodes available for fine-tuning to the same number of episodes available for training MaxIteration. For more details we refer to Appendix C.

**Experimental Results**   Figure 3 contains a set of demonstrative results. The full results are deferred to Appendix C. The selected results in Figure 3 highlight three properties of MaxIteration:

1. There are cases where max-following not only increases the return but actually leads to solving a task successfully even when none of the constituent policies achieve success.
2. With successful constituent policies, max-following can significantly increase the success rate.
3. max-following can sometimes increase return but not necessarily lead to success demonstrating the need to better understand which attributes make up good constituent policies in the future.

The results in Appendix C demonstrate that in all cases, MaxIteration is at least as good as the best constituent policy which is not the case for algorithms from prior work [Liu et al., 2023] as discussed earlier. Moreover, MaxIteration consistently leads to greater return improvement than fine-tuning given the same amount of data. Fine-tuning with substantially more resources would

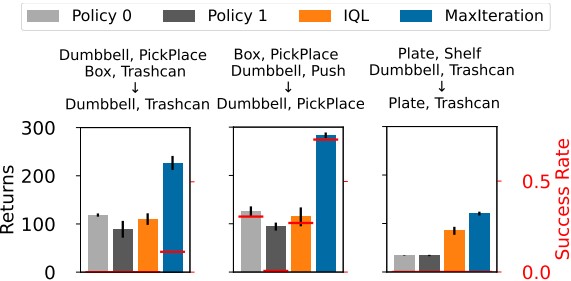

Figure 3: Policies 0 and 1 correspond to the pre-trained policies using IQL on the intial tasks above the arrow in each graph. That is, in the left most subfigure, Policy 0 corresponds to the policy of picking and placing a dumbbell, whereas Policy 1 corresponds to the policy of moving a box into the trashcan. Mean return and success rate over 5 seeds of MaxIteration compared to fine-tuning IQL on selected tasks. Error-bars correspond to standard error. Full bars correspond to returns and red lines indicate the success rate of each algorithm. MaxIteration can yield improvements in return but increased return does not always yield success.

eventually surpass the performance of MaxIteration as MaxIteration is limited to competing with the max-following benchmark which can be suboptimal.

## 6 Conclusion

We introduce MaxIteration, an algorithm to efficiently learn a policy that is competitive with the approximate max-following benchmark (and hence also with all constituent policies). We provide empirical evidence that max-following utilizing skill-learning enables us to learn how to complete tasks that it would be inefficient to learn from scratch, but that are superior to other individually trained experts for fixed given skills.

**Limitations and Future Work**    Our goal in this work has been to learn a policy that competes with an approximate max-following policy under minimal assumptions. However, we still assume efficient batch learnability of constituent value functions, which will not always be feasible in practice. While it seems likely that our oracle assumption is necessary for learning an approximate max-following policy, we leave proving this claim for future work. We also leave consideration of alternative ensembling approaches to future work. Max-value ensembling is sensitive to slight differences in the values between constituent policies whereas, e.g., softmax takes into account the relative 'weighting' of values. In addition, it would be interesting to characterize the amount of improvement we can obtain over our constituent policies or prove conditions under which our approximate max-following policy is competitive with a true max-following policy or the optimal policy. One could also extend this analysis to ensembling methods like softmax and study the nature of guarantees in that setting. Extending beyond MDPs to the partially observable setting, and to the discounted infinite-horizon setting, would also add richness to the class of problems we could consider.

## Acknowledgments and Disclosure of Funding

The authors are partially supported by ARO grant W911NF2010080, DARPA grant HR001123S0011, the Simons Foundation Collaboration on Algorithmic Fairness, and NSF grants FAI-2147212 and CCF-2217062.

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

## A  MDP Examples

### A.1  LQR max-following parametric class vs. constituent policies

$$\min_{\{u_t\}_{t=0}^{\infty}} \quad \sum_{t=0}^{\infty} \gamma^t (x_t^T Q x_t + u_t^T R u_t)$$
$$\text{subject to} \quad x_{t+1} = A x_t + B u_t + w_t,$$

To motivate the use of max-following policies in a richer class of MDPs, we consider a traditional control problem with continuous state and action spaces: the discrete linear quadratic regulator. Note that here we analyze the infinite horizon discounted case so that we can analyze the time-invariant value function, but episodic analogues exist. Consider the following setting where $\gamma \in [0,1]$ is a discount factor, and $w_t \sim \mathcal{N}(\mathbf{0}, \sigma^2 I)$. Here, we consider the simple case where $Q, R, A = I$ and $B = (1+\epsilon)I$. We know that the optimal policy is of the form $u = -K^* x$ [Bertsekas, 2012] and we set two policies that are only stable along one component and unstable along the other of the form $u_1 = -K_1 x$ and $u_2 = -K_2 x$. It is important to note that the value functions of the individual policies and the optimal policies have exact quadratic forms like $V(x) = x^T P x + q$, but the max-following policy is not necessarily within the same parametric class. For example, $P_1$ is the solution to the Lyapunov equation $P_1 = (I + K_1^T K_1 + \gamma(A - K_1)^T P_1(A - K_1))$ and $q_1 = \frac{\gamma}{1-\gamma}\sigma^2 \operatorname{tr}(P_1)$. A similar formula exists for policy 2.

In LQR, for the $K_1, K_2$ controllers described above, a max-following policy is able to attain higher value than the individual expert policies that have an unstable direction in one axis. Moreover, we see that the optimal policy is obviously superior to all the other policies, but that a max-following policy is more competitive with it than the other individual expert policies. A max-following policy is ultimately able to benefit from the stabilizing component of each axis of the individual policies, which ultimately lets it perform better than any given individual one.

## B  Additional Proofs

**Lemma 4.1** (Worst approximate max-following policy competes with best fixed policy). *For any $\varepsilon \in (0,1]$ and any episode length $H$, let $\beta \in \Theta(\frac{\varepsilon}{H})$. Then for any MDP $\mathcal{M}$ with starting state distribution $\mu_0$, and any $K$ policies $\Pi^k$ defined on $\mathcal{M}$,*

$$\min_{\pi \in \Pi_{\beta}^{k*}} \mathbb{E}_{s_0 \sim \mu_0}\left[V^{\hat{\pi}}(s_0)\right] \geq \max_{k \in [K]} \mathbb{E}_{s_0 \sim \mu_0}\left[V^k(s_0)\right] - O(\varepsilon).$$

*Proof.* We will prove the claim inductively, showing that for all $C \in [H]$, if we run any approximate max-following policy for $C$ steps, and then continue following the policy $\pi^k$ chosen at step $C$ for the rest of the episode, then our expected return is not much worse than if we had followed any fixed $\pi^k$ for the whole episode.

Somewhat more formally, recalling the definition of the set of approximate max-following policies $\Pi_{\beta}^{k*}$ (Definition 2.3), at every time $h \in [H]$ and state $s \in \mathcal{S}$, a policy $\pi \in \Pi_{\beta}^{k*}$ takes action $\pi_h^t(s)$ for a $\pi^t \in \Pi^k$ such that $V_h^t(s) \geq \max_{k \in [K]} V_h^k(s) - \beta$. Letting $\pi^{t(s,h)}$ denote the $\pi^t \in \Pi^k$ that $\pi$ follows at state $s$ and time $h$, we will show that if at some step $C \in [H]$ we have

$$\mathbb{E}_{s_0 \sim \mu_0, P}\left[\sum_{h=0}^{C} R(s_h, \pi_h(s_h)) + \sum_{h=C+1}^{H-1} R(s_h, \pi_h^{t(s_C,C)}(s_h))\right] \geq \max_{k \in [K]} \mathbb{E}_{s_0 \sim \mu_0}\left[V^k(s_0)\right] - O(\tfrac{\varepsilon(C+1)}{H}),$$

for all $\pi \in \Pi_{\beta}^{k*}$, then the same holds for $C+1$ for all $\pi$.

In the base case, $C = 0$, the claim

$$\mathbb{E}_{s_0 \sim \mu_0, P}\left[\sum_{h=0}^{H-1} R(s_h, \pi_h^{t(s_0,0)}(s_h))\right] \geq \max_{k \in [K]} \mathbb{E}_{s_0 \sim \mu_0}\left[V^k(s_0)\right] - O(\tfrac{\varepsilon}{H})$$

for all $\pi \in \Pi_\beta^{k^*}$ and all $\pi^k \in \Pi^k$, follows straightforwardly from the definition of $\Pi_\beta^{k^*}$ and setting of $\beta \in \Theta(\frac{\varepsilon}{H})$, since

$$\mathbb{E}_{s_0 \sim \mu_0, P}\left[\sum_{h=0}^{H-1} R(s_h, \pi_h^{t(s_0,0)}(s_h))\right] = \mathbb{E}_{s_0 \sim \mu_0}[V^{\pi^{t(s_0,0)}}(s_0)]$$

$$\geq \mathbb{E}_{s_0 \sim \mu_0}\left[\max_{k \in [K]} V^k(s_0) - O(\tfrac{\varepsilon}{H})\right]$$

$$\geq \max_{k \in [K]} \mathbb{E}_{s_0 \sim \mu_0}\left[V^k(s_0)\right] - O(\tfrac{\varepsilon}{H}).$$

We now prove the inductive step. We wish to show that if at step $C$, we have for some $\pi \in \Pi_\beta^{k^*}$

$$\mathbb{E}_{s_0 \sim \mu_0, P}\left[\sum_{h=0}^{C} R(s_h, \pi_h(s_h)) + \sum_{h=C+1}^{H-1} R(s_h, \pi_h^{t(s_C,C)}(s_h))\right] \geq \max_{k \in [K]} \mathbb{E}_{s_0 \sim \mu_0}\left[V^k(s)\right] - O(\tfrac{\varepsilon(C+1)}{H}),$$

then continuing to follow $\pi$ at step $C+1$ and following $\pi^{t(s_{C+1}, C+1)}$ thereafter reduces expected return by $O(\frac{\varepsilon}{H})$. Now if $\pi_{C+1}(s_{C+1}) = \pi_{C+1}^t(s_{C+1})$ for $\pi^t \in \Pi^k$, it must be the case that

$$V_{C+1}^t(s_{C+1}) \geq \max_{k \in [K]} V_{C+1}^k(s_{C+1}) - O(\tfrac{\varepsilon}{H}),$$

otherwise $\pi \notin \Pi_\beta^{k^*}$. It follows that

$$\mathbb{E}_{s_0 \sim \mu_0, P}\left[\sum_{h=0}^{C+1} R(s_h, \pi_h(s_h)) + \sum_{h=C+2}^{H-1} R(s_h, \pi_h^{t(s_{C+1}, C+1)}(s_h))\right]$$

$$= \mathbb{E}_{s_0 \sim \mu_0, P}\left[\sum_{h=0}^{C} R(s_h, \pi_h(s_h)) + V_{C+1}^{t(s_{C+1}, C+1)}(s_{C+1})\right] \quad \text{(by definition of } V \text{ and } \pi_{C+1}(s_{C+1}))$$

$$\geq \mathbb{E}_{s_0 \sim \mu_0, P}\left[\sum_{h=0}^{C} R(s_h, \pi_h(s_h)) + \max_{k \in [K]} V_{C+1}^k(s_{C+1}) - O(\tfrac{\varepsilon}{H})\right] \quad \text{(from } \pi \in \Pi_\beta^{k^*})$$

$$\geq \mathbb{E}_{s_0 \sim \mu_0, P}\left[\sum_{h=0}^{C} R(s_h, \pi_h(s_h)) + V_{C+1}^{t(s_C,C)}(s_{C+1}) - O(\tfrac{\varepsilon}{H})\right]$$

$$= \mathbb{E}_{s_0 \sim \mu_0, P}\left[\sum_{h=0}^{C} R(s_h, \pi_h(s_h)) + \sum_{h=C+1}^{H-1} R(s_h, \pi_h^{t(s_C,C)}(s_h))\right] - O(\tfrac{\varepsilon}{H}) \quad \text{(by definition of } V)$$

$$\geq \max_{k \in [K]} \mathbb{E}_{s_0 \sim \mu_0}\left[V^k(s)\right] - O(\tfrac{\varepsilon(C+2)}{H}) \quad \text{(by inductive hypothesis)}$$

and so the claim holds for time $C+1$, for any $\pi \in \Pi_\beta^{k^*}$ for which it holds for time $C$. We showed the base case $C=0$ hold for all $\pi \in \Pi_\beta^{k^*}$, and therefore we have

$$\mathbb{E}_{s_0 \sim \mu_0, P}\left[\sum_{h=0}^{C} R(s_h, \pi_h(s_h)) + \sum_{h=C+1}^{H-1} R(s_h, \pi_h^{t(s_C,C)}(s_h))\right] \geq \max_{k \in [K]} \mathbb{E}_{s_0 \sim \mu_0}\left[V^k(s)\right] - O(\tfrac{\varepsilon(C+1)}{H})$$

for all $C \in [H]$. In particular, for $C = H - 1$ we conclude that

$$\mathbb{E}_{s_0 \sim \mu_0, P}\left[\sum_{h=0}^{C} R(s_h, \pi_h(s_h))\right] \geq \max_{k \in [K]} \mathbb{E}_{s_0 \sim \mu_0}\left[V^k(s)\right] - O(\varepsilon)$$

and it follows that

$$\min_{\pi \in \Pi_\beta^{k^*}} \mathbb{E}_{s_0 \sim \mu_0}\left[V^{\hat{\pi}}(s_0)\right] \geq \max_{k \in [K]} \mathbb{E}_{s_0 \sim \mu_0}\left[V^k(s_0)\right] - O(\varepsilon).$$

$\square$

# C   Additional information about experiments

For our experiments, we use a heuristic version of MaxIteration that operates in rounds. First, the algorithm collects a set of trajectories using every policy to initialize the respective value functions. Then, in every round the algorithm for every policy exectues the max-following policy for $\beta$ steps and the switches to the respective constituent policy. At the end of each round, value functions of constituent policies are updated. $\beta$ is uniformly spaced along the full horizon and thus, depends on the number of rounds and the horizon. The total number of episodes is an upper bound on the number of samples collected which is what we determine to compare run-times between MaxIteration and IQL. Finally, we use a $\gamma$ discounting which has been shown to have regularizing effects on the value function updates [Amit et al., 2020].

For IQL, we use the d3rlpy implementations [Seno and Imai, 2022] and code provided by Hussing et al. [2024].

## C.1   Hyperparameters

Both algorithms are run for $10,000$ steps initially (to initialize value functions for MaxIteration and to pre-fill the buffer for IQL) before doing updates and then for $50,000$ steps for online training.

All neural networks use ReLU [Glorot et al., 2011] Multi-layer perceptrons with 2 layers and a hidden dimension of 256 per layer.

Table 1: Hyperparameters for MaxIteration

| Optimizer | Adam |
|---|---|
| Adam $\beta_1$ | 0.9 |
| Adam $\beta_2$ | 0.999 |
| Adam $\varepsilon$ | $1e-8$ |
| Value Function Learning Rate | $1e-4$ |
| Number of rounds | 50 |
| Number of gradient steps per round | 40,000 |
| Batch Size | 64 |
| $\gamma$ | 0.99 |

Table 2: Hyperparameters for Implicit Q-Learning

| Optimizer | Adam |
|---|---|
| Adam $\beta_1$ | 0.9 |
| Adam $\beta_2$ | 0.999 |
| Adam $\varepsilon$ | $1e-8$ |
| Actor Learning Rate | $4e-3$ |
| Critic Learning Rate | $4e-3$ |
| Batch Size | #Tasks $\times 256$ |
| n_steps | 1 |
| $\gamma$ | 0.99 |
| $\tau$ | 0.005 |
| n_critics | 2 |
| expectile | 0.7 |
| weight_temp | 3.0 |
| max_weight | 100 |

## C.2 Full results on CompoSuite

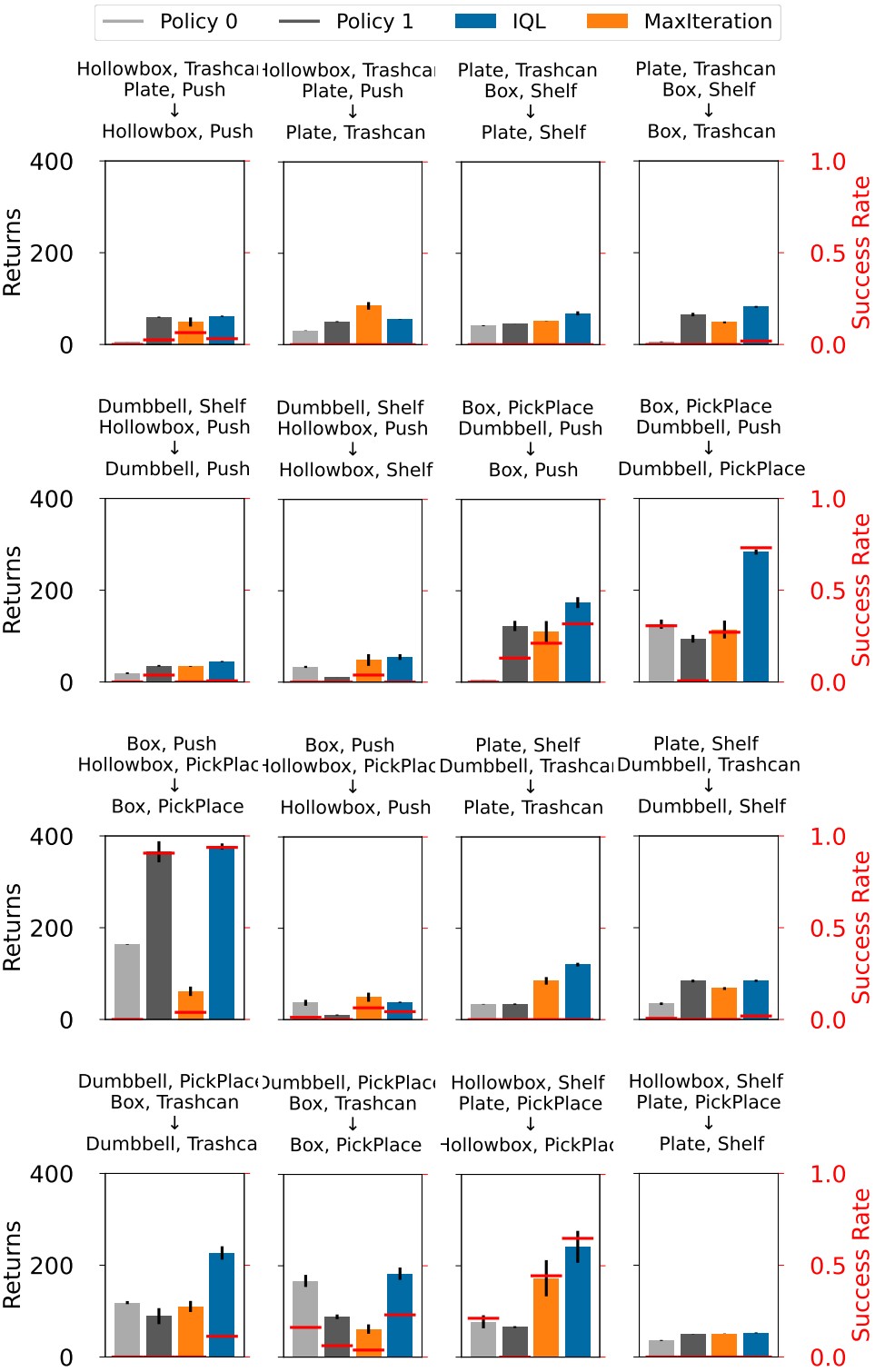

Figure 4

## C.3 Results on DM Control

We run our MaxIteration algorithm on the DM Control benchmarks [Tunyasuvunakool et al., 2020] similar to the MAPS [Liu et al., 2023] setup. In their setup, the constituent policies correspond to different 3 checkpointed models in one run of the online Soft-Actor critic [Haarnoja et al., 2018] algorithm. As a result, it is generally true that the latest checkpointed model will outperform the previous two checkpoints meaning one constituent policy is strictly better everywhere than the others. We report the final performance over 5 seeds using 16 evaluation trajectories in Figure 5. The results show that our algorithm behaves as expected and always uses the best oracle. Without policy improvement operator, this setup does not allow us to exceed the performance of the constituent policies.

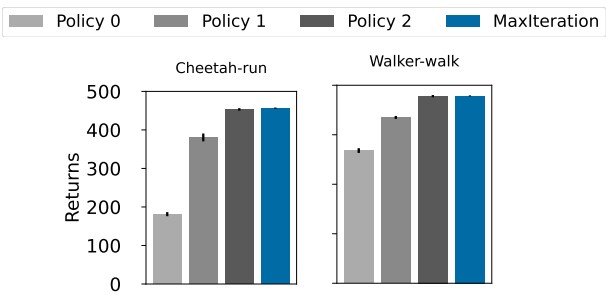

Figure 5: Mean return over 5 seeds of MaxIteration on DM Control tasks [Tunyasuvunakool et al., 2020]. Error-bars correspond to standard error. MaxIteration always selects the best performing constituent policy.

## C.4 Computational Resources

Our experiments were conducted using a total of 17 GPUs including both server-grade (e.g., NVIDIA RTX A6000s) and consumer-grade (e.g., NVIDIA RTX 3090) GPUs. Training the constituent policies from offline data takes less than 2 hours. Our MaxIteration algorithm takes about 3 hours to train while the baseline fine-tuning takes around 1 hour. A large chunk of the runtime cost stems from executing the simulator.

