# OpenReview forum: "Oracle-Efficient Reinforcement Learning for Max Value Ensembles"
_NeurIPS.cc/2024/Conference — NeurIPS 2024 poster_

### Official Review · Reviewer_Qu2b · 2024-07-01

**Soundness:** 3
**Presentation:** 2
**Contribution:** 3
**Rating:** 6
**Confidence:** 3

**Summary:**

The paper considers the setting in which several good policies are available for some Markov Decision Process, and the agent has to learn how to combine them in a way that allows to achieve higher performance than following a single of the constituent policies. This problem is quite wide and a large number of methods exist to combine policies, with varying assumptions and properties. This paper proposes a method that can realistically be implemented, by considering that per-policy value functions can be learned for an ever-increasing horizon $h$, and then the action to execute in the environment is the one executed by the policy that has the largest value in the current state and current horizon.

The novel aspect of the contribution seems to be the reliance on $h$ and the iterative nature of the algorithm. In MDPs with a finite time horizon, we have a finite amount of $h$ values, so a finite amount of value functions to learn. Combined with the fixed amount of constituent policies, this leads to an algorithm whose complexity scales well even to very large state-spaces, because the size of the state space does not intervene in the compute requirements of the algorithm.

The proposed method is discussed from a theoretical perspective, and promising empirical results are provided. The experiments consider complicated robotic tasks, with an interesting setting and motivation for this work: the constituent policies are almost-optimal policies for various tasks, and the new policy to learn from a combination of the existing ones is learned against a new task. So, this work seems to be applicable to multi-task RL.

**Strengths:**

The proposed method seems sound, easily to implement (provided that the paper is made clearer) and to lead to impressive empirical results. The assumptions and limitations of the approach are well-discussed, which helps deciding whether it would apply well on some specific problem.

**Weaknesses:**

While the contribution seems of high quality, the paper lacks clarity and intuition, which may make it difficult to reproduce.

- Examples of oracles for the value functions should be given, to better indicate to people whether the oracle can be a value function learned on another task, or obtained from rollouts, or requires a simulator. The oracle is an important part of the algorithm, as it is queried $K$ times per horizon step and its output is directly used to perform an argmax operation (the result of the oracle does not seem to be distilled to some learned function)
- The paper should be a bit more explicit about how to produce the $\mu_h$ distributions and the fact that this requires resetting the environment and performing actions in it. Not every environment is resettable at will by the agent, and executing all these actions requires an online setting.
- The core of the paper is the use of approximate max-following policies, defined in Definition 2.3. The definition is very dry and the reader has to carefully look at the notations to understand where everything comes from. For such an important definition, an intuition and maybe an example would have been very useful. Later in the paper (Figure 2), examples of environments and corresponding max-following policies are used in some argument, but without explaining what a max-following policy is. Thus, the definition is very dry, yet at the core of important arguments in the paper.

**Questions:**

Given the average-low clarity of the paper, I may have mis-understood several parts of the contribution. I expect the authors to disagree with some of the remarks written above. I would welcome to be corrected, and for the authors to take the opportunity to improve the paper given the possible ways it can be mis-understood.

**Limitations:**

The paper does not seem to have potential negative societal issues, and its scientific limitations are well-discussed.

---

> ### Author Rebuttal · Authors · 2024-08-06
>
> Dear Reviewer Qu$2$b, thank you so much for highlighting the scaling dependence of our algorithm/its ability to work in large state spaces well and for relating it to our empirical results and the broader multi-task RL area.
>
> **Weaknesses**
> > Examples of oracles for the value functions should be given
>
> In practice, the oracle can be implemented using an arbitrary function approximator such as a neural network. The input distribution of the oracle is approximated by rolling out the max-following policy to some time step $h$ and then rolling out a constituent policy up to the horizon of the environment. This can happen either in a simulator or on a real-world system. The collected data is then used to fit the neural network. The learned neural network is the output of the oracle. See Appendix C for additional detail on the procedure. For additional formal clarification of the oracle, see response to weakness $2$.
>
> >The paper should be a bit more explicit about how to produce the
>  distributions and the fact that this requires resetting the environment and performing actions in it.
>
> We thank the reviewer for pointing out the lack of clarity in how the $\mu_h$ distributions are defined and sampled. We provide some explanation below, and will add clarification to the paper as well.
>
> - We first want to highlight that sampling from $\mu_h$ does not require that the agent can reset the environment at will. We only require what is typically required in the episodic setting -- that the agent explores for an episode of $H$ steps, where $H$ is finite and fixed across all of training. After these $H$ steps, the agent is then reset to a state sampled from the distribution over starting states.
> - The distributions $\mu_h$ are (informally) defined as follows: at iteration $h \in [H]$ of our algorithm, the agent has already learned a good approximate max-following policy for the first $h$ steps of the episode. The distribution $\mu_h$ is the distribution over states visited by the agent at step $h$ if it begins from a state drawn from the starting state distribution and then follows the approximate max-following policy it has learned thus far for $h$ steps.
> - That means to sample from $\mu_h$, the oracle can simply run the approximate max-following policy for $h$ steps to arrive at a state $s_h$, which is a sample from $\mu_h$. It can then do whatever it likes for the remainder of the episode, and so does not need to reset at arbitrary time steps.
> - In practice, since the oracle needs to produce a good approximation of the value function $V_h^k$ at time $h$ for policy $\pi^k$ on states sampled from $\mu_h$, we should think of it as using the remainder of the episode to obtain an unbiased estimate of the expectation of $V_h^k$ on the distribution $\mu_h$. That is, once it has sampled a state $s_h$ by running the approximate max-following policy for $h$ steps, it just executes policy $\pi^k$ for the remainder of the episode. The accumulated reward obtained by following policy $\pi^k$ from state $s_h$ for steps $h$ through $H$ gives the oracle an unbiased estimate of $\mathbb{E}_{s_h \sim \mu_h}[V^k_h(s_h)]$. To implement our oracle assumption, we could use many such unbiased estimates as training data to train a neural network, to learn a good approximate value function for $\pi^k$ at time $h$ on distribution $\mu_h$.
>
> >The core of the paper is the use of approximate max-following policies, defined in Definition 2.3. The definition is very dry and the reader has to carefully look at the notations to understand where everything comes from.
>
> Thank you for the feedback on the readability of Definition 2.3. First, as we start explaining in line 134 and following, a "max-following policy is defined as a policy that at every step follows the action of the [constituent] policy with the highest value in that state." With **approximate** max-following policies one can run into issues arising from the fact that value functions are not perfectly accurate anymore. We will add some additional motivating intuition leading up to the technical definition 2.3 similar to that of Definition 2.1. The purpose of our observations in section $4$ is to provide the intuition for max-following and approximate max-following. For instance, imagine a scenario where two of the value functions are very close to each other for some state. Now, due to noise introduced from approximation it might look as if the function with the true lower value actually has the higher of the two values (see Observation 4.6). As outlined in section $4$, accounting for this noise in our benchmark requires comparing the learned policy to a class of policies (the class of approximate max-following policies) rather than a single baseline policy, and so the technicality of Definition 2.3 is unfortunately necessary. If the reviewer thinks it would improve readability, we would be happy to move Section 4 before Section 3, where we present our algorithm, so that the reader can have these examples in mind before we present our main results.
>
> **Questions**
>
> We are extremely grateful for this reviewer's thorough feedback and attention to detail and we hope to be able to incorporate much of the advice provided to build a better paper. We hope to be able to use the discussion period to clarify some of the details surrounding our oracle assumption both in theory and in practice and work on improving the exposition/clarity of our definitions and examples to help create a better paper.

---

### Official Review · Reviewer_Z6cc · 2024-07-12

**Soundness:** 3
**Presentation:** 2
**Contribution:** 3
**Rating:** 5
**Confidence:** 3

**Summary:**

This paper presents an approach to enabling Reinforcement Learning (RL) by improving the process of generating an optimal policy. Their method assumes that there is access to a particular class of Markov Decision Process (MDP). The intuition is having a collection of bases determined as constituent policies. Under some theoretically supported assumptions, the policy learning process they propose guarantees consistently obtaining a policy that is at least as good as the best constituent policy and potentially better. The method relies on an algorithm that generates policies from the max-following class. The authors tested their framework with a practical version of such an algorithm (MaxIteration) in 16 robotic tasks. An essential aspect of their method is that it relies on heuristics to create the collection of constituent policies.

**Strengths:**

This paper presents a structured theory consolidated over the definition of a policy function class called max-following. The authors provided a practical algorithm that can generate policies belonging to that class. They proved their proficiency through robotic tasks, compared to an offline RL method called Implicit Q-Learning (IQL).

**Weaknesses:**

This class falls under a set of observations provided by the authors that are necessary for the theory to hold. Also, the bases of constituent policies rely on heuristics. That could restrict the potential application of their training method.

I couldn't find the explanation of Policy 0 and Policy 1 in Figure 3. This figure could be more apparent, as understanding your baselines is essential.

**Questions:**

- What are Policy 0 and Policy 1 in Figure 1?
- How do you explain the tasks where all the methodologies had a very low success rate?

**Limitations:**

The authors addressed their method's main limitation: the need for an oracle for the methodology to hold. They claimed that they will address this concern in the future.

---

> ### Author Rebuttal · Authors · 2024-08-06
>
> Dear Reviewer Z$6$cc, thank you so much for your feedback on our paper and for highlighting the practical nature of our theoretically-grounded algorithm.
>
> **Weaknesses**
> > We are unsure what "this class falls under a set of observations provided by the authors that are necessary for the theory to hold” means. Would you be willing to clarify a bit more?
>
> One interpretation of this statement may be that our observations from section $4$ are necessary for our theory to hold. This would be incorrect as we simply provide the observations to clarify the properties of our max-following and approximate max-following approach. These are \textit{not} assumptions needed in order for our algorithm and theoretical results to hold, and are rather meant for our reader to gain intuition into some of the definitions we provided. The main assumption our paper makes is access to an ERM oracle.
>
> > It is also unclear to us what it means that the bases of our constituent policies rely on heuristics. We would appreciate clarification on this point as well.
>
> The constituent policies are not chosen based on heuristics, and are rather chosen based upon access to some pre-existing class of policies. We do not make assumptions about the properties of these policies, beyond the assumption that a regression oracle is able to give us reasonable approximations to their value functions. We hope that in many realistic settings there are policies that are pre-trained such that they are skilled in one domain but not necessarily trained to be good at other tasks. Thus, we can compose them well, but skill-specific pre-training is not necessary for our theoretical results to hold.
>
>   >  I couldn't find the explanation of Policy 0 and Policy 1 in Figure 3. This figure could be more apparent, as understanding your baselines is essential.
>
> We appreciate this pointer. For the camera ready version, we will update the caption of the Figure to include a description of the policies.
> Policies $0$ and $1$ correspond to the pre-trained policies using IQL on the intial tasks above the arrow in each graph. That is, in the left most subfigure of Figure $3$, Policy $0$ corresponds to the policy of picking and placing a dumbbell, whereas Policy $1$ corresponds to the policy of moving a box into the trashcan. The MaxIteration algorithm enables the robot move a dumbbell into the trashcan without needing the robot to train separately on that task by reusing the earlier existing policies.
>
> **Questions**
>
> > What are Policy 0 and Policy 1 in Figure 1?
>
> In our figure $1a$, the policies correspond to moving left or right and in $1b$ correspond to moving right, left, or up. If the review is referring to Figure $3$, we refer to the previous response.
>
> > How do you explain the tasks where all the methodologies had a very low success rate?
>
> Note that in the experimental setting, the initial policies (i.e. policies 0 and 1) are pre-trained to solve tasks distinct from the task that we test them on. Thus, we expect the pre-trained policies to be bad at the test task. Our algorithm only guarantees that we are at least as good as the best individual policy. In some cases, we see that combining policies is not sufficient to achieve success. That is because the tasks we chose are robotics tasks with highly complex dynamics and the dynamics of moving a plate versus a dumbbell using a robotic arm differ quite a bit. In such cases, simply switching between the two policies is insufficient to solve the task. However, this opens up interesting directions for future work such as including minor policy update steps to the learning process without requiring state-space dependence.

---

> > ### Comment · Reviewer_Z6cc · 2024-08-13
> > **Answer to Rebuttal**
> >
> > Dear authors, thanks for your efforts in clarifying my comments. Here are some clarifications to the questions from my side that didn't appear very clear:
> >
> > > We are unsure what "this class falls under a set of observations provided by the authors that are necessary for the theory to hold” means. Would you be willing to clarify a bit more?
> >
> > I think you understood what I meant, and thanks to your answer, I can now identify that the observations are not necessary conditions (assumptions) on which to base your theoretical results.
> >
> > > It is also unclear to us what it means that the bases of our constituent policies rely on heuristics. We would appreciate clarification on this point as well.
> >
> > This part comes from the experimental results that require you to use heuristic-based versions of the algorithm. Thank you for clarifying that this is only related to this part and is not required in the theoretical proof. The confusing part comes from the abstract, where you stated: "One line of research [...] the natural assumption that we are given a collection of heuristic base or constituent policies [...]." I believe this part makes the reader assume that the constituent policies come from heuristics, which is why I commented in the first place. Maybe you can adjust the abstract to clarify the difference.
> >
> > > Regarding Figure 3
> >
> > Thank you for explaining and considering my feedback to clarify what Policies 0 and 1 mean. I think improving the details you provided about them would help make your experiments more understandable. Your clarification about this was very helpful in understanding the underperforming tasks in my last question.
> >
> > Thanks for taking the time to answer all of them; besides what I stated above, I don't have additional comments.

---

> > > ### Author Response · Authors · 2024-08-13
> > >
> > > Dear reviewer Z6cc, we are grateful for your feedback and will make several changes in the next iteration of the manuscript.
> > > * We had hoped to convey that simple heuristic policies can sometimes be useful even when they are not complex, but we see now that there is ambiguity in this statement. We will adjust the abstract as you suggested and point out in the main text that constituent policies can, but must not necessarily, be heuristic.
> > > * We will also make changes to the caption and text with respect to the description of the policies 0 and 1 including a paragraph similar to what we provided in the rebuttal.
> > >
> > > Thank you for engaging in this discussion phase, we greatly appreciate it. We hope you are now more positively disposed to our paper and are happy to discuss further if there are any other points of confusion

---

### Official Review · Reviewer_BR1J · 2024-07-15

**Soundness:** 3
**Presentation:** 2
**Contribution:** 3
**Rating:** 5
**Confidence:** 2

**Summary:**

The paper presents an algorithm called MaxIteration for addressing the challenges of RL in large or infinite state spaces. The core idea is to compete with a max-following policy, which at each state selects the action of the constituent policy with the highest value. The MaxIteration algorithm is efficient, requiring only access to an empirical risk minimization (ERM) oracle for value function approximation of the constituent policies, without needing the value functions themselves.

**Strengths:**

1. The algorithm is computationally efficient, scaling well with large state spaces.
2. The paper provides a solid theoretical foundation with proofs of the algorithm's effectiveness.
3. It improves upon existing policies without needing to explore the entire state space.
4. The algorithm's performance is validated by experiments on robotic simulation tasks.

**Weaknesses:**

1. The algorithm assumes access to an ERM oracle, which might not be practical in all scenarios.
2. While its efficiency in simulation tasks, the algorithm might be complex to implement in real-world systems.

**Questions:**

1. Are there any specific cases where the algorithm's performance might degrade?
2. How does the algorithm deal with non-stationary environments or changing dynamics?

**Limitations:**

Please see the above comments.

---

> ### Author Rebuttal · Authors · 2024-08-06
>
> Dear Reviewer BR1J, thank you for your feedback on our work and highlighting the value of our theoretically motivated empirical algorithm.
>
> **Weaknesses**
>    > The algorithm assumes access to an ERM oracle, which might not be practical in all scenarios.
>
> Many machine learning problems are known to be computationally hard in the worst case, but Empirical Risk Minimization (ERM) has still proven to be a useful framework in practice. For instance, the machine learning community has made great strides in solving supervised learning problems using neural networks over the past 10 years.
> Our oracle assumption can be thought of as providing us the guarantee that we can learn an approximate value function well in such a batch-ERM setting. That means, we are reducing the problem of learning an approximate max-following policy to a simpler supervised learning problem over a given distribution. In other words, our goal with this work is to provide guarantees under the assumption that in practice ERM is easy and neural networks do what neural networks do.
>
>  > While its efficiency in simulation tasks, the algorithm might be complex to implement in real-world systems.
>
> We agree that one of the main challenges of implementing RL algorithms on real-world systems is the required sample complexity. However, we would like to highlight that our work is an attempt at reducing the required number of samples to obtain good policies. Our experimental results use only around $80$ trajectories while common on-policy RL algorithms require several thousand [Mendez et al. 2022].
>
> **Questions**
>    >  Are there any specific cases where the algorithm's performance might degrade?
>
> We provably cannot do worse than the constituent policies with our approach which gives us a baseline for our algorithm's worst-case performance. Intuitively, we can think of it this way: if it is worse to switch between policies, we can always resort to using only a single constituent policy. Exactly for this reason, as we experimentally show, there are cases where MaxIteration also cannot do drastically better and ultimately performs similarly to one of the base constituent policies themselves. This is also the case in Figure $1b$.
>
>    >  How does the algorithm deal with non-stationary environments or changing dynamics?
>
> This is a very intriguing question and we thank the reviewer for bringing it up. Seeing what happens when the performance of a constituent policy changes due to changes in the environment is a very interesting idea. In general, we believe that it is possible to obtain efficient routines for such settings. However, this would require redefining the setting as well as some of the other definitions (and likely changing the algorithm). For the current manuscript we believe that this question is out of scope but it is an excellent direction for future work we would like to pursue.

---

### Author Rebuttal · Authors · 2024-08-06

First and foremost, we would like to thank all the reviewers for their time and feedback on our paper. We thank reviewers BR1J and Z6cc for highlighting the usefulness of our theoretically motivated but practically employable algorithm. We also thank reviewer Qu2b for highlighting the simplicity of our algorithm and the strong words about our empirical results.


Given the common questions around our oracle assumption, we would like to make a clarifying statement up front. Many machine learning problems are known to be computationally hard in the worst case, but Empirical Risk Minimization (ERM) has still proven to be a useful framework in practice. For instance, the machine learning community has made great strides in solving supervised learning problems using neural networks over the past 10 years.
Our oracle assumption can be thought of as providing us the guarantee that we can learn an approximate value function well in such a batch-ERM setting. That means, we are reducing the problem of learning an approximate max-following policy to a simpler supervised learning problem over a given distribution. In other words, our goal with this work is to provide guarantees under the assumption that in practice ERM is easy and neural networks do what neural networks do.

---

### Decision · Program_Chairs · 2024-09-25

**Decision:**

Accept (poster)

**Comment:**

This paper considers the setting in which several good policies are available for some MDP, and an agent has to learn how to combine them to achieve higher performance than following a single of the constituent policies. The core idea behind the proposed algorithm is to use with a max-following policy, which at each state selects the action of the constituent policy with the highest value. The algorithm's efficiency is demonstrated both theoretically and empirically.